

# Microclimate mapping using novel radiative transfer modelling

Florian Zellweger[1], Eric Sulmoni[1], Johanna T. Malle[1], Andri Baltensweiler[1], Tobias Jonas[2],
Niklaus E Zimmermann[1], Christian Ginzler[1], Dirk Nikolaus Karger[1], Pieter De Frenne[3], David
Frey[1], Clare Webster[1,2,4]

[1]Swiss Federal Research Institute WSL, Birmensdorf, Switzerland

[2]WSL Institute for Snow and Avalanche Research SLF, Davos Dorf, Switzerland

[3]Forest and Nature Lab, Department of Environment, Faculty of Bioscience Engineering, Ghent University, Ghent,
Belgium

[4]Department of Geosciences, University of Oslo, Norway

*Correspondence to*: Florian Zellweger (florian.zellweger@wsl.ch)





**Abstract**

Climate data matching the scales at which organisms experience climatic conditions are often missing. Yet, such
data on microclimatic conditions are required to better understand climate change impacts on biodiversity and
ecosystem functioning. Here we combine a network of microclimate temperature measurements across different
habitats and vertical heights with a novel radiative transfer model to map daily temperatures during the vegetation
period at 10 meter spatial resolution across Switzerland. Our data reveals strong horizontal and vertical variability
in microclimate temperature, particularly for maximum temperatures at 5 cm above the ground and within the
topsoil. Compared to macroclimate conditions as measured by weather stations outside forests, diurnal air and
topsoil temperature ranges inside forests were reduced by up to 3.0 and 7.8 °C, respectively, while below trees
outside forests, e.g. in hedges and below solitary trees, this buffering effect was 1.8 and 7.2 °C. We also found that
in open grasslands, maximum temperatures at 5 cm above ground are on average 3.4 °C warmer than that of
macroclimate, suggesting that in such habitats heat exposure close to the ground is often underestimated when
using macroclimatic data. Spatial interpolation was achieved by using a hybrid approach based on linear mixed
effects models with input from detailed radiation estimates that account for topographic and vegetation shading,
as well as other predictor variables related to the macroclimate, topography and vegetation height. After accounting
for macroclimate effects, microclimate patterns were primarily driven by radiation, with particularly strong effects
on maximum temperatures. Results from spatial block cross-validation revealed predictive accuracies as measured
by RSME's ranging from 1.18 to 3.43 °C, with minimum temperatures generally being predicted more accurately
than maximum temperatures. The microclimate mapping methodology presented here enables a more biologically
relevant perspective when analysing climate-species interactions, which is expected to lead to a better
understanding of biotic and ecosystem responses to climate and land use change.

**Keywords**

Biodiversity, Climate Change, Forest structure, LiDAR, Microclimate modelling, Radiative transfer model,
Remote sensing, Topography



## 1 Introduction

Current understanding of climate and climate change impacts on biodiversity and ecosystem functioning are often based on macroclimate data available at spatial scales much coarser than the microclimatic conditions experienced by organisms (Bramer et al., 2018; Potter et al., 2013). Most of these macroclimate datasets are based on interpolations of standardised weather station data, typically using temperature measurements taken outside of forests and above grasslands at ~2 m above ground level. However, across landscapes, local topography and vegetation cover create heterogeneous microclimates through altering local radiation regimes, air mixing and evapotranspiration. Macroclimate data are therefore limited in representing near-surface microclimate conditions close to or in the ground and under vegetation canopies where most terrestrial organisms reside. Given the importance of microclimates for the physiology of organisms as well as for key ecosystem processes such as carbon, nutrient and water cycling, accurately predicting microclimates at high spatial and temporal resolutions is fundamental for understanding climate change impacts on biodiversity and ecosystem functioning (Jones, 2014; De Frenne et al., 2021).

Variation in microclimate is driven by the topography, vegetation, soil, and the water balance, all of which modulate near-surface temperatures from the prevailing macro-scale meteorological conditions (Geiger et al., 2009). Local controls on microclimates include the buffering of forest understories against macroclimate temperature extremes (De Frenne et al., 2019; Chen et al., 1999) and the high heterogeneity of surface microclimates in topographically complex environments, such as mountains (Scherrer and Körner, 2010). For example, maximum temperatures and temperature extremes can be reduced in areas shaded by topography and/or vegetation due to the reduction in incoming shortwave solar radiation, an effect that can be increased by evapotranspirative cooling if water availability is not limited (De Frenne et al., 2021). Minimum temperatures, on the other hand, are modulated by factors such as heat retention by vegetation canopies through reduced outgoing longwave radiation and reduced wind speeds, as well as cold air flow and pooling in topographic depressions, particularly during the night and calm atmospheric conditions (Dobrowski, 2011; Geiger et al., 2009).

Fortunately, mapping of microclimates has recently been facilitated by advanced microclimate measuring and modelling techniques (Maclean et al., 2018; Zellweger et al., 2019a; Maclean et al., 2021) and the compilation of large databases of in-situ microclimate measurements (Lembrechts et al., 2020). These new data streams and technologies are now being used to create large scale microclimate datasets and mapping products that will contribute to a better understanding of the climate-related distribution and functioning of organisms (Lembrechts et al., 2019b; Suggitt et al., 2018; Maclean and Early, 2023; Haesen et al., 2023; Lembrechts et al., 2021).

Mapping microclimate across landscapes has particularly been assisted by remote sensing technologies such as Light Detection and Ranging (LiDAR) and digital photogrammetry, which provide detailed information about the topography and vegetation structure that can be used as input variables to model near-surface temperatures (Jucker et al., 2018; Frey et al., 2016; Duffy et al., 2021; Greiser et al., 2018; Maclean et al., 2018). A remaining challenge in microclimate mapping is incorporating radiation transfer through vegetation canopies, which currently is often crudely represented via the use of canopy cover and density proxies such as Leaf Area Index (LAI), canopy height and/or canopy cover. These proxies lack the directional component to radiation transfer and typically generalise the canopy away from individual tree-level structure, both of which impact the physiology of organisms. Using



these proxies can therefore lead to errors in estimates of canopy transmissivity in heterogeneous forest canopies (Musselman et al., 2013), thereby increasing uncertainties for analysing microclimate effects on plant species composition (Zellweger et al., 2019b). Newer radiative transfer models, based on remotely sensed 3D vegetation structure datasets are now able to accurately calculate canopy transmissivity maps at meter- and even submeter-scale resolution by directly accounting for detailed and realistic canopy structure in relation to the changing daily and seasonal solar position (Musselman et al., 2013; Bode et al., 2014; Tymen et al., 2017; Webster et al., 2020; Kükenbrink et al., 2021; Webster et al., 2023). The increasing availability of 3D vegetation structure datasets at large spatial extents enables including detailed radiative transfer variables in microclimate mapping approaches, incorporating tree-level vegetation structure across landscapes and habitats.

A further limitation to current microclimate analysis and mapping is the lack of reliable in-situ microclimate measurements across a wide range of habitats. In places exposed to sunlight, for example, many commonly used microclimate temperature loggers - shielded or unshielded - record biased measurements due to radiative fluxes operating on the thermometer (Maclean et al., 2021). Fortunately, these biases can now be minimised by using ultra fine-wire thermocouples with a low thermal emissivity and highly reflective surface, recording accurate estimates of air temperatures even in places exposed to sunlight or close to the ground (Maclean et al., 2021). Deploying these measurement devices across multiple habitat types that span wide ranges of variation in vegetation structure and topography is thus required to arrive at a reliable reference dataset that is representative of the entire spectrum of microclimate conditions within environmentally heterogeneous regions. This would, for example, allow researchers to include the often ignored but specific thermal conditions beneath trees outside forests, e.g. in hedges providing important habitats and increase habitat connectivity, in microclimate mapping products (Vanneste et al., 2020).

Here we combine a state-of-the-art radiative transfer model with a comprehensive microclimate measurement network to infer and map daily microclimate temperatures at three vertical heights and 10 m spatial resolution across the whole of Switzerland. The resulting microclimate dataset is a major step forward towards taking a realistic organism's perspective when studying species-climate interactions and will be relevant to many fields of biological and environmental sciences, including fundamental and applied ecology, hydrology, agriculture and forestry (De Frenne et al., 2021; Bramer et al., 2018).

## 2 Materials and Methods

### 2.1 Study area

This study was carried out in Switzerland, which covers 41,248 km$^2$ of Central Europe. Mountains cover c. 70% of the country, lowlands the remaining 30%. One-third of the land is forested, with a larger proportion in the mountain areas. The forest composition consists of coniferous (42%), mixed (34%) and deciduous (24%) forests (Brändli et al., 2020). 4% of the country (1,813 km$^2$) are covered by trees outside of forests, e.g. trees found in hedges or solitary trees (Malkoç et al., 2021).

### 2.2 Temperature measurements inside and outside forests

We implemented a nationwide network of microclimate temperature sensors following a hierarchical stratified sampling design. First, we identified eight regions to represent the main macroclimate gradients in Switzerland (Fig. 1). These regions align with the long-term Forest Ecosystem Research (LWF) network, covering gradients



ranging from the lowlands with a temperate, relatively warm climate to higher and cooler elevations receiving more precipitation, to inner alpine regions with a continental climate and regions in the southern Alps with an insubric climate. In each region, we installed temperature sensors in several plots, covering the regional variation in forest structure and topography. Inside forests, we identified locations with low to high topographic slope angles and topographic positions, as well as locations with different slope orientations, i.e. from north to south facing slopes. In each of the forest locations, we sampled one plot with high and one plot with low canopy cover, as visually estimated in the field. All forest plots were at least 50 m away from the nearest forest edge. Outside forests we sampled grasslands with different slope orientations, as well as high and low relative topographic positions, i.e. ridges to valley bottoms. Finally, in each region we selected plots below trees outside forests. In hedge type habitats, i.e., linear accumulations of woody vegetation, we placed the loggers in the middle of each hedge. Below solitary trees we placed the loggers at half the distance between the tree trunk and outer crown projection line. Due to regional plot availability and suitability as determined by field visits, the number of final plots per region varied, ranging from 6 to 17 (median of 15) plots per region. The total number of plots was 107, with 62 plots in forests, 22 below trees outside forests, and 23 in open grasslands. Due to logistic reasons only forest plots were sampled in the Pfynwald and southern Ticino, thereby increasing the coverage of different forest types in our sample. As a result, our sample plots represented the observed range of environmental conditions across the study area well, as indicated by a comparison between sampled and observed predictor variable space across the area used for making predictions (Appendix B, Table B1).

In each plot, microclimate temperatures were measured at 1 m and 5 cm above the ground surface, as well as below the ground in the topsoil at 5 cm depth. These heights were chosen because we expect a large degree of vertical temperature variation between these heights, as indicated by common temperature profiles (De Frenne et al., 2021) and because these heights are representative of the strata in which many organisms reside (e.g. herbaceous plants, tree seedlings, ground arthropods, soil fungi and bacteria). We acknowledge that sampling entire vertical forest profiles reaching the top tree canopy would be desired from an ecological viewpoint, but we were not able to achieve this due to logistic reasons. Above ground air temperatures (both at 1 m and 5 cm) were recorded hourly using Lascar Electronics EL-USB-TC loggers with unshielded ultra-fine wire thermocouples (0.08 mm) taped to a 1 m tall fencing pole (Fig. 1d-e). For sampling at 5 cm below the ground we used standard Lascar EL-USB-1 loggers placed into a buried small sealable plastic tube, also recording at an hourly resolution. Both the thermocouples and standard Lascar logger types have a measuring accuracy of 0.3 °C, as reported by Lascar Electronics. To check if the measurements of the unshielded thermocouples were affected by direct sunlight we performed an experimental sensitivity test, which revealed no significant effect of direct sunlight (Appendix A). The measurement period started on 8th of June 2021 and ended 31st of October 2022, with slightly varying starting dates per region as determined by the site visits to install all loggers. The sampling duration was thus long enough to include a wide range of weather conditions, from wet and cold to hot and dry periods. All sites were revisited every two to three months for maintenance and to retrieve the data. Together with careful checks and corrections for obvious outliers and data artefacts introduced by device malfunction or disturbance by animals, this maintenance enabled us to build up a mostly seamless time series of hourly temperature data, with an overall loss of data of less than 5 %. Each site was georeferenced using a Trimble® GeoExplorer 6000 with an accuracy after post-processing of c. 1 m.



For the analysis presented here we pooled all data collected between June 2021 to October 2021 and April 2022 to October 2022, broadly representing the longest vegetation period observed across Switzerland. We further excluded all temperature recordings that were made under snow cover, which mainly affected the 5 cm above ground and 5 cm below ground measurements at high elevations. We did this because snow blankets introduce spatial and temporal variability of atmospheric decoupling in temperatures below or within a snow blanket, and

this variability cannot accurately be modelled with our predictor variables.

We calculated temperature offsets to analyse the variation of microclimate in comparison to the macroclimate, i.e., we deducted the microclimate temperature from the macroclimate temperature (see section predictor variables for details). To build the final time series dataset for the spatial modelling we aggregated the hourly data to daily maximum ($T_{max\ micro}$), mean ($T_{mean\ micro}$) and minimum ($T_{min\ micro}$) temperature. $T_{max\ micro}$ was defined as the 24-hour

95th percentile and $T_{min\ micro}$ as the 5th percentile. $T_{mean\ micro}$ is the arithmetic daily mean temperature (with n = 24). These three daily temperature statistics are the dependent variables for the models used to predict nationwide microclimate temperature maps as outlined below.

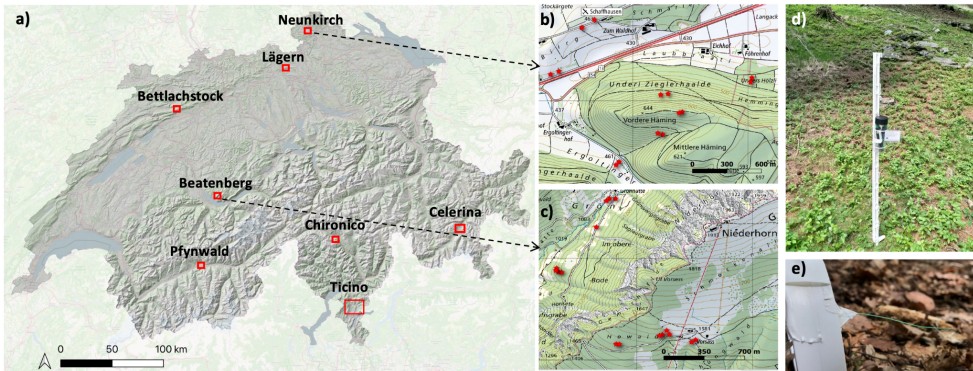

**Figure 1**. Sampling design for microclimate measuring network across Switzerland. a) distributions of the eight
regions spanning a wide macroclimatic gradient across the country. In each region, sites were identified to represent regional variation in vegetation structure and topography (see text for details); b) and c) distribution of sites in the regions of Neunkirch and Beatenberg, respectively. d) installation of microclimate sensors, with an ultra-fine wire thermocouple measuring the temperature 5 cm above the ground as shown in e).

### 2.3 Predictor variables

The development of our predictor variable set was guided by the assumption that the variation of near surface microclimate temperatures as measured by our sensor network is strongly related to variation in macroclimate temperature, followed by effects of local-scale variation in topography, vegetation structure, and associated radiation regimes.

To derive the macroclimate we used interpolated daily maximum $T_{max\ Macro}$, daily mean $T_{mean\ Macro}$ and daily
minimum $T_{min\ Macro}$ data from meteorological stations as provided by MeteoSwiss at a 1 km² nationwide grid (Frei, 2014). The underlying data for these macroclimate layers were collected at 131 weather stations at 2 m height above ground, in open localities outside forests across the country. For model fitting as well as for the final



predictions we needed to downscale the 1 km$^2$ macroclimate data to our 10 m target resolution, which was guided
by the resolution of our other predictor variables, especially those describing the topography and vegetation height

as described below. To this end we applied a lapse rate correction to the macroclimate layers, which is important
in our mountainous study area, where the pronounced altitudinal gradients and related lapse rates cause large
temperature differences within an original 1 km$^2$ grid cell. Daily lapse rates were calculated for each $T_{max\ macro}$,
$T_{mean\ macro}$ and $T_{min\ macro}$ separately, using a moving window regression based on the respective 1 km$^2$ temperature
grid cell and the corresponding 1 km$^2$ MeteoSwiss reference DTM with a moving 3 x 3 window size. We thus

estimated, for each window, how much the temperature changes as a function of the regional elevation gradient.
The resulting regression estimate is the daily lapse rate $\Gamma$ in °C/m. Only grid cells with regression results with an
$R^2 > 0.85$ were considered to ensure a reliable lapse rate value. Remaining empty raster cells were filled in a second
step with the average lapse rate within a moving window of 5x5 cells. Based on visual inspection of the resulting
lapse rate maps, this combination of window sizes resulted in the best achievable estimation of locally prevailing

lapse rate values. The result of this process was nationwide 1 km$^2$ resolution auxiliary maps of daily lapse rates
for each $T_{max\ macro}$, $T_{mean\ macro}$ and $T_{min\ macro}$. The processing workflow is illustrated in Appendix F, Fig. F1.

For the downscaling to the target resolution, the 1 km grids of both the lapse rate- and the 1 km$^2$ MeteoSwiss $T_{max}$
$_{macro}$, $T_{mean\ macro}$ and $T_{min\ macro}$ rasters were first resampled to 10 m resolution with bilinear interpolation. Then, the
10 m MeteoSwiss daily $T_{max\ macro}$, $T_{mean\ macro}$ and $T_{min\ macro}$ maps were corrected for sub-grid elevation variability

using the 10 m lapse rate information as follows:

$$\Delta z = z_{DTM} - z_{MeteoSwiss} \tag{1}$$

$$T_{max\ macro\ cor} = T_{max\ macro\ org} + \Delta z * \Gamma_{Tmax\ macro} \tag{2}$$

$$T_{mean\ macro\ cor} = T_{mean\ macro\ org} + \Delta z * \Gamma_{Tmean\ macro} \tag{3}$$

$$T_{min\ macro\ cor} = T_{min\ macro\ org} + \Delta z * \Gamma_{Tmin\ macro} \tag{4}$$

where $\Delta z$ is the difference between each 10 m grid cell of Swissalti3D DTM elevation $z_{DTM}$ (swisstopo, 2020) and
the nearest 1km grid cell of the MeteoSwiss reference elevation for the temperature grids $z_{MeteoSwiss}$. $T_{max\ macro\ org}$,
$T_{mean\ macro\ org}$ and $T_{min\ macro\ org}$ are the resampled 10 m MeteoSwiss temperature rasters and $\Gamma$ the respective
temperature lapse rates. The resulting lapse rate corrected daily macroclimate air-temperature maps at 10 m
resolution, i.e. $T_{max\ macro\ cor}$, $T_{mean\ macro\ cor}$ and $T_{min\ macro\ cor}$ were used as predictor variables for the modelling and



mapping of microclimate temperatures as measured within our network of microclimate temperature loggers (c.f. Section on Microclimate Modelling below).

We also tested for effects of daily cloud cover, by incorporating actual macroclimate global radiation as derived from MeteoSwiss and found that daily cloud cover did not improve the predictive performance of our models, possibly because daily macroclimate temperatures already incorporate daily weather effects.

### 2.3.1 Radiative transfer model: Shortwave transmissivity, sky-view fraction and subcanopy radiation

Small-scale variability in radiation within forests was represented by accounting for explicit tree-level forest structure around each point. Variability in the diffuse shortwave and longwave radiation components were represented using the 180° sky-view fraction ($V_f$, also known as diffuse transmissivity). Variability in the direct shortwave component was accounted for by determining the proportion of the solar disc obscured by vegetation

or topography (also known as time-varying direct-beam transmissivity, $\tau_{dir}$), which varies both in space and in time as the solar position changes in the sky. Direct-beam transmissivity and sky-view fraction were both calculated using the model CanRad (Webster et al., 2023), which uses synthetic 180° hemispherical images to replicate the topography and vegetation as seen by the ground/plant surface (Fig. 2). The radiation transfer model simulations represent only leaf-on conditions, which implies - contrary to direct-beam transmissivity - that skyview

fraction varies spatially but is temporally static.  To resolve the fine-scale temporal variability of direct-beam transmissivity we calculated it at 2-minute intervals and then averaged it to hourly time steps. CanRad was run at the point scale at 20 m intervals across the entire domain, totalling 87,795,419 points and 265,320 time steps across the annual solar cycle.

At all points, terrain shading was included by using 5 m and 25 m DTMs (swisstopo, 2020). The 5 m DTM was

included up to 300 m radius from each point to represent local terrain variability around each model point. The coarser 25 m DTM was used up to a 10 km radius from each point to calculate the topographic horizon line, accounting for terrain shading from nearby mountains.

For the above radiative transfer modelling we used the module C2R (CanopyHeightModel2Radiation) within CanRad which achieves a realistic representation of the overhead canopy structure based on a canopy height model

(CHM) to determine the geometric arrangement of vegetation surrounding a point with information on forest type and subsequent leaf area of individual tree crowns. The CHM was available at 1 m resolution based on lidar datasets ranging from 1-30 points/m² acquired across Switzerland from 2012-2021, with one region (i.e. Pfynwald) having older data from 2003. The high spatial resolution of the CHM across the model domain ensured the effect of individual trees on ground-surface shading were explicitly incorporated. Forest type information was provided

by the nationwide forest mix rate dataset from Waser *et al. (*2017) to discriminate between deciduous and evergreen forest types and the Swiss forest ecoregions dataset (FOEN, 2022) was used to distinguish between needleleaf or broadleaf forest types. For a more thorough description of the radiative transfer modelling, see (Webster et al., 2023).




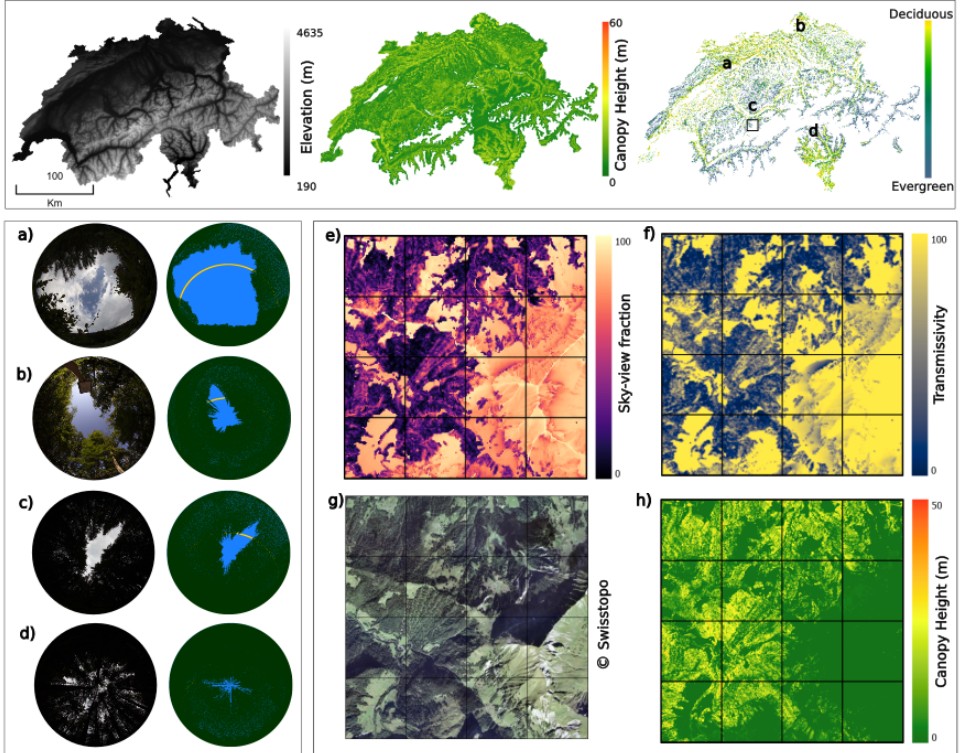

**Figure 2:** Top: nationwide input datasets used in CanRad to calculate synthetic hemispherical images: elevation (left), canopy height (middle), forest type mix rate and locations of outputs shown in bottom panels (right). Bottom left: examples of hemispherical photographs and corresponding synthetic images at the same locations; a) below shrubs; b) deciduous broadleaf forest; c) northern alpine evergreen coniferous forest; d) southern alpine evergreen coniferous forest; in the synthetic images the yellow line corresponds to the solar track on June 22. Sky-view fraction is then calculated as the ratio based on a non-linear weighting of the blue + yellow area relative to the total area. Bottom right: Example of model estimates of skyview fraction (e) and average transmissivity for June (f) over a 4 km$^2$ region in the central alps, as indicated by the black square in top right forest mix rate map. Aerial image (g) and canopy height (h) shown for context. Note: photographs in a-d have not been corrected for lens distortion compared to synthetic images which have an equiangular lens projection.

The hourly estimates of direct-beam transmissivity were aggregated to daily values by averaging the values each day between the hours of 9AM and 4PM. The assumption here was that the daily maximum microclimate temperature is mostly dependent on solar radiation within this time interval. These daily aggregates were averaged to monthly averages thereafter and resampled from 20 m to 10 m resolution using bilinear interpolation. Finally, we multiplied these monthly average transmissivity values with monthly averages of daily clear sky direct shortwave irradiation as estimated by (Zimmermann and Roberts, 2001), yielding what hereafter is referred to as direct radiation proxy. Note that this proxy represents both the spatial and the seasonal variation in subcanopy direct shortwave radiation.



As an additional predictor variable, we also extracted vegetation height at 10 m resolution from the above-
mentioned CHM.

### 2.3.2 Topographic position- and wetness index

We used the swissalti3D DTM with a 10 m resolution to derive indices of topographic position (TPI) and wetness
(TWI). TPI and TWI serve as indicators for cold air flow and pooling, as well as exposure to wind, thus affecting
near surface temperatures (Ashcroft and Gollan, 2013; Daly et al., 2008). TPI, or relative elevation, is defined as
the normalised difference between the elevation of a focal cell and the average elevation within a minimum radius
(here zero) and a maximum radius (here 120 meters). TWI describes the lateral water flow and was calculated as
follows:

$$TWI = ln \frac{a}{tan\ b} \tag{5}$$

where $a$ is the local upslope catchment area and $tan\ b$ is the local slope in radians (Freeman, 1991).

### 2.3.3 Soil moisture and rain

Soil moisture has been shown to affect near surface temperatures, for example by lowering the buffering magnitude
of forest floor temperatures compared to outside forest temperature under dry conditions (von Arx et al., 2013).
To account for potential effects of soil moisture and precipitation on our measured microclimate temperatures we
calculated a predictor variable termed "rain sum", using daily precipitation data from MeteoSwiss on a 1 km grid.
To calculate this variable for each day (i.e. at a daily resolution) we summed up the precipitation over the preceding
30 days, giving a linearly decreasing weight to values further in the past.

**Table 1**. Predictor variables used for modelling, with the range of values representing the variable ranges across
our microclimate sampling plots.

| Variable name | Description | Range (mean) | Unit |
|---|---|---|---|
| Macroclimate temperature | Daily maximum near surface (2 m) lapse rate corrected air-temperature ($T_{max\ macro\ cor}$) as derived from MeteoSwiss | -5.6 - 37.0 (17.8) | °C |
| | Daily mean near surface (2 m) lapse rate corrected air-temperature ($T_{mean\ macro\ cor}$) as derived from MeteoSwiss | -7.6 - 28.6 (13.4) | °C |
| | Daily minimum near surface (2 m) lapse rate corrected air-temperature ($T_{min\ macro\ cor}$) as derived from MeteoSwiss | -13.4 - 23.1 (8.6) | °C |
| Direct radiation proxy | Proxy for average daily direct clear sky shortwave irradiation on the ground and beneath vegetation canopies | 0 - 29410 (8600) | kJ/m2/day |
| Skyview fraction | Proportion of sky visible taking an upward perspective on the ground and beneath vegetation canopies | 2 - 98 (41.5) | % |
| Vegetation height | Vegetation height derived from canopy height model | 0 - 35.1 (13.3) | m |
| Rain sum | Weighted sum of daily precipitation amount over preceding 30 days | 2.6 - 320.1 (62.9) | mm |
| Topographic position | Relative topographic position describing the plot elevation in | -1.0 - 0.97 (- | Index |



| | | | |
|---|---|---|---|
| | relationship to the surround- ing elevations. Valley bottoms have low values; elevated locations, such as ridges, have high values | 0.04) | |
| Topographic wetness | Topographic wetness index representing the lateral water flow | 1.1 - 9.6 (3.2) | Index |
| Northness | Cosine of topographic aspect. Northness is a continuous variable describing the topographic exposition ranging from completely north ex- posed to completely south exposed | -1 - 1 (-0.1) | Index |
| Slope | Topographic slope | 0.3 - 41.7 (17.9) | Degrees |

### 2.4 Microclimate modelling

We statistically related the plot-level measurements, i.e., the daily $T_{max\ micro}$, $T_{mean\ micro}$ and $T_{min\ micro}$ measurements
at different vertical heights, to the spatial predictor variables (Table 1), and used the resulting model equations to predict national maps of daily microclimate over the entire gridded domain. The sample sizes for the 1 m, 5 cm and the topsoil datasets were n = 33'390, n = 30'781 and n = 27'662, respectively. As mentioned above, please note that our dependent variables were the actual microclimate temperature measurements and not the temperature offsets. We tested three modelling approaches to analyse the predictive performance of our predictor variables.

First, we fitted linear mixed effects models with our predictor variables as fixed effects and "region" as a random intercept term to account for the non-independence among replicates from the same region, using restricted maximum likelihood in the *lmer* function from the lme4 package in R (Bates et al., 2015). All variables were standardised, i.e., rescaled to have a mean of zero and a standard deviation of one, to increase the interpretability of relative effect sizes among the predictor variables. For $T_{max\ micro}$ and $T_{mean\ micro}$ we included the interaction term
between the macroclimate temperature and the direct radiation proxy at ground level and below canopy, as it has been shown that the maximum temperature buffering capacity of tree canopies can increase with warmer temperatures (De Frenne et al., 2019). $T_{min\ micro}$ was modelled as a function of sky-view fraction instead of direct radiation, to account for the negative net longwave radiation during night as a presumed main driver of $T_{min\ micro}$.

The second approach was a random forest regression model, using the "*randomForest*" package in R (Liaw and
Wiener, 2002). Two variables were randomly sampled as candidates at each split, with a total number of 500 trees. We used this machine learning algorithm because it automatically considers variable interactions and non-linear relationships between dependent and independent variables. These features may lead to increased predictive accuracy because such interactions and non-linear relationships may indeed be present in our data, as it has been shown that effects of vegetation structure and topography on near surface temperatures may be non-linear
(Zellweger et al., 2019c).

To further test for non-linear responses, we also used general additive mixed-effects models (GAMMs) as our third modelling approach, applying the *gamm* function in the "mgcv" package in R (Wood, 2017). We again added "region" as a random term and used REML as the smoothing parameter estimation method for the model.

To evaluate the predictive performance of our models we applied a spatial block cross validation approach (Roberts
et al., 2017). We therefore iteratively used data from 7 out of the 8 regions for model fitting and predicted the data from the left-out 8th region to compare the predicted with the observed values.



As indicated in the results section, we used the linear mixed effects models to produce daily microclimate maps across Switzerland at 10 m resolution covering the period between 1st of April and 31st of October, for all the years between 2012 to 2021. We calculated these maps for daily $T_{max\ micro}$, $T_{mean\ micro}$ and $T_{min\ micro}$, each at 1 m and

5 cm above ground, as well as in the topsoil 5 cm below ground. As noted, these data are representative for snow-free conditions during the vegetation period and leaf-on conditions. The ten-year period has been chosen acknowledging that changes in tree cover and density do occur, where most of our LiDAR data used for the radiation modelling was acquired during the years 2012-2021. Because we have neither sampled microclimate data in urban areas nor settlements, nor in non-vegetated areas, such as scree or glacial habitats, we masked those

areas out from our microclimate maps, using the land cover mapping product Vector25 (swisstopo, 2022).

### 3 Results

#### 3.1 Temperature offsets in different habitats

The temperature offsets describe the differences between the microclimate and macroclimate data and thus indicate microclimate variation not captured in macroclimate data. Our nationwide sampling in different habitats revealed

strong horizontal and vertical variability in temperatures during the vegetation period, with a particularly high degree of variation in daily maximum near surface and topsoil temperature measured at 5 cm above ground and 5 cm below ground, respectively (Fig. 4). Daily temperature extremes as measured by $T_{max\ micro}$ and $T_{min\ micro}$ were considerably reduced in the topsoil and in forests, as indicated by negative offset values for $T_{max\ micro}$ and positive offset values of $T_{min\ micro}$.

In forests, the daily $T_{max\ micro}$ were cooler than the daily $T_{max\ macro}$ outside forests, with mean offset values of -1.3 and -1.1 °C for air temperature at 1 m and 5 cm above ground, respectively, and -5.2 °C in the topsoil (Appendix C, Table C1). Daily $T_{min\ micro}$ were generally warmer, with average offset values of 1.8, 1.7, and 2.6 °C at 1 m, 5 cm above ground and in the topsoil, respectively. In forests, the resulting absolute difference between Tmax and Tmin offsets, i.e. the total temperature buffering effect, were thus 3.0, 2.9, and 7.8 °C, respectively.

Below trees outside forests we also found reduced daily extreme temperatures as compared to $T_{max\ macro}$, but the magnitude of the temperature buffering effect was lower than in forests. Daily $T_{max\ micro}$ were on average lower by -0.7, -0.2 and -4.9 °C at 1 m, 5 cm above ground and in the topsoil, respectively, while daily $T_{min\ micro}$ were on average higher by 1.1, 0.9 and 2.3 °C. The resulting total temperature buffering effect below trees outside forests was thus 1.8, 1.1, and 7.2 °C, respectively.

Unlike in forests and trees outside forests, temperature offsets for maximum air temperatures at 5 cm in grasslands were found to be positive, i.e., 3.4 °C, indicating that near surface $T_{max\ micro}$ in open habitats are often underestimated when using macroclimate data. Moreover, topsoil $T_{min\ micro}$ in open grasslands were warmer than the macroclimate by 3.5 °C on average. Across all habitat types we found that the degree of variation in offset values, particularly $T_{max\ micro}$ offset values, was greater at 5 cm above ground than at 1 m or in the topsoil,

suggesting a high spatial variability in near surface air temperatures across and within the different habitat types.



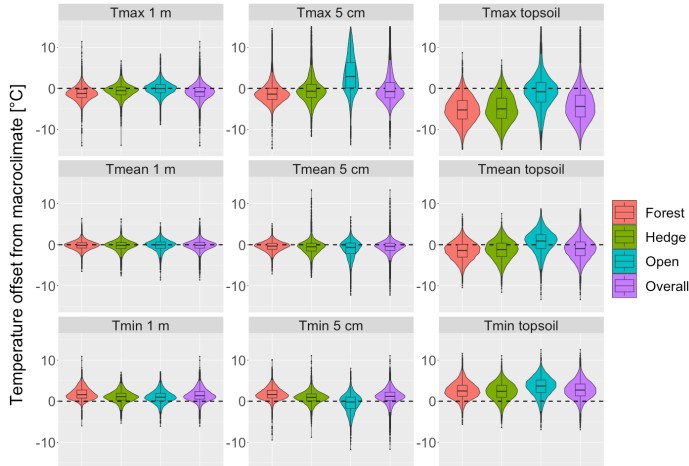

**Figure 3.** Offsets between macroclimate and in situ measured microclimate temperature during April to October per habitat type and overall data points. The offsets were calculated by subtracting the microclimate temperature from the lapse rate corrected macroclimate temperature ($T_{macro}$- $T_{micro}$). Negative offset values thus indicate cooler microclimates compared to the macroclimate, and vice-versa.

### 3.2 Model performance

The predictive performance of our models ranged between $R^2$ values of 0.54 and 0.95, and root mean squared errors (RMSE) 1.2 and 3.4 °C (Table 2, Fig. 4). Microclimate temperatures at 1 m height were predicted with the highest accuracy, followed by temperatures at 5 cm and in the topsoil. $T_{min\ micro}$ at 5 cm and in the topsoil were predicted considerably more accurately than the respective $T_{max\ micro}$ values. We also found large ranges in predicted temperature offsets, with generally wider ranges for $T_{max\ micro}$ than for $T_{min\ micro}$. The widest range of offsets were found for topsoil $T_{max\ micro}$, followed by $T_{max\ micro}$ at 5 cm and 1 m.

The observed patterns in model performance and predicted offset value ranges were broadly similar across the three tested modelling approaches but it is noteworthy that random forests as well as GAMMs predicted considerably larger offset ranges (Appendix D, Table D1). Yet, when evaluated based on block cross validation, linear mixed effects models had the highest overall predictive skill. We thus used the linear mixed effects models to evaluate individual predictor variable effects and to calculate the final microclimate maps.





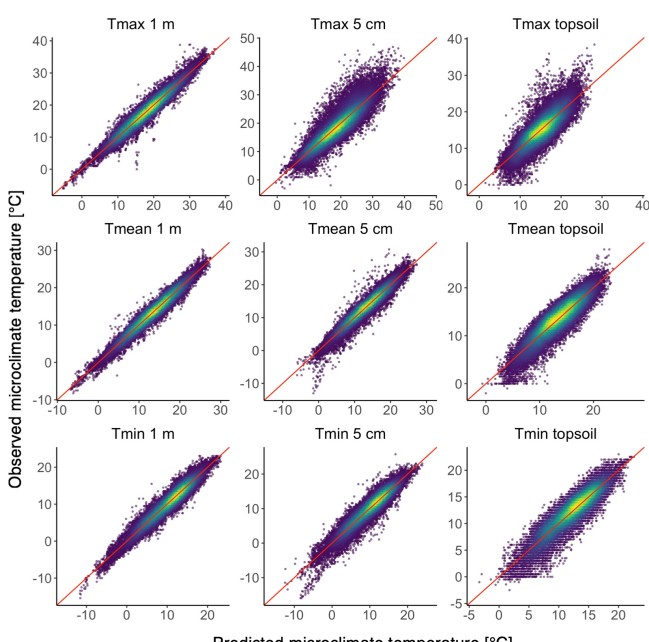


**Figure 4.** Predicted versus observed plots showing predictions from linear mixed effects models. The sample density is indicated by the color scale, with yellow showing highest sample densities; the red line represents the 1:1 relationship.

**Table 2**: Predictive performance of linear mixed effects models, quantified using block cross validation. We also
report the predicted offset ranges, i.e., minimum to maximum, and mean in brackets, calculated as the difference between the predicted microclimate and the macroclimate.

| | Temperature 1 m above ground | | | Temperature 5 cm above ground | | | Temperature in topsoil 5 cm below ground | | |
|---|---|---|---|---|---|---|---|---|---|
| | R2 | RMSE | Predicted offset range | R2 | RMSE | Predicted offset range | R2 | RMSE | Predicted offset range |
| $T_{max\ micro}$ | 0.92 | 1.7 | -3.0 - 1.4 (-0.6) | 0.73 | 3.4 | -5.6 - 7.0 (0.4) | 0.54 | 3.1 | -15.4 - 8.0 (-4.0) |
| $T_{mean\ micro}$ | 0.95 | 1.2 | -2.4 - 1.9 (-0.1) | 0.9 | 1.6 | -3.3 - 3.6 (-0.4) | 0.75 | 2.0 | -7.5 - 9.1 (-0.9) |
| $T_{min\ micro}$ | 0.92 | 1.5 | -0.8 - 3.4 (1.5) | 0.88 | 1.8 | -1.3 - 4.3 (1.2) | 0.78 | 1.9 | -1.5 - 9.5 (2.9) |

### 3.3 Predictor variable effects

In line with the expectation that microclimate patterns broadly follow macroclimate dynamics we found that
macroclimate variables had the strongest effects on the microclimate, as indicated by the highest standardised variable estimates (Fig. 5). Yet, most of the predictor variables related to radiation, vegetation and topography significantly modulated the local variation of microclimate.

We found that $T_{mean\ micro}$ and $T_{max\ micro}$ were strongly related to direct radiation, with particularly large effects on $T_{max\ micro}$ at 5 cm above ground and in the topsoil 5 cm below ground. The interaction effects between the



macroclimate and direct radiation were relatively weak but significant. Specifically, the effect of direct radiation increased with increasing $T_{max\ macro}$ for $T_{max\ micro}$ at 1 m and 5 cm, but the opposite was true for $T_{mean\ micro}$. Skyview fraction strongly modulated $T_{min\ micro}$, with negative effects on $T_{min\ micro}$ at 1 m and 5 cm, and positive effects on topsoil $T_{min\ micro}$. Vegetation height had the largest effects on $T_{max}$, cooling down $T_{max\ micro}$ as vegetation height increased across all three measurement heights. Higher water availability as estimated by the rain sum of the

preceding thirty days generally had a small but significant cooling effect on microclimate temperatures at 1 m and 5 cm, and a warming effect on topsoil temperatures. From all topography variables tested, topographic wetness and northness had the largest effects, predominantly cooling temperatures across all three heights at higher levels of topographic wetness and northness. In general, radiation and the vegetation height affected microclimate temperatures more strongly than variables related to water content and topography.

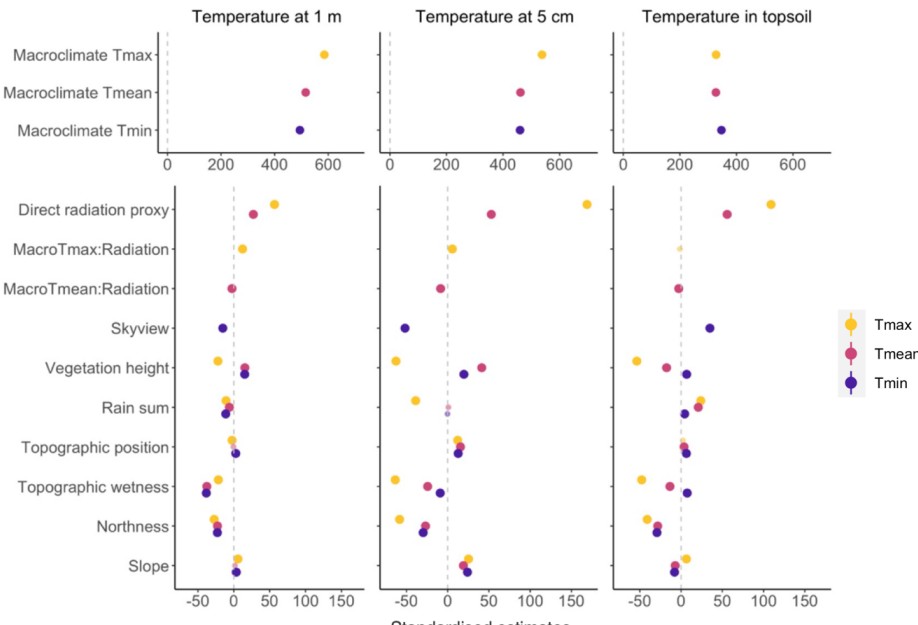


**Figure 5**. Standardised model coefficient estimates from linear mixed effects models for daily microclimate temperatures at different vertical heights, i.e., at 1 m and 5 cm above ground and 5 cm below ground. The estimates on the x-axis indicate the standardised effect sizes and direction of each variable. Standard errors are indicated by error bars, however due to the large sample size the error bars are small and invisible. Small transparent dots

indicate non-significant ($p > 0.05$) relationships.

### 3.4 Microclimate maps

Our microclimate maps show pronounced differences compared to currently available macroclimate layers (Fig. 6). Spatial variation in microclimates is particularly evident between forest and non-forest areas, and microclimate effects of trees outside of forests, e.g., in hedges or similar linear tree habitats, become visible. The strongest

vertical temperature differences emerge between topsoil and near surface air temperatures. All daily maps of $T_{min\ micro}$, $T_{mean\ micro}$ and $T_{max\ micro}$, for all three vertical heights, have also been aggregated to monthly averages, which



are publicly available (see data access section). The broad coverage in our model calibration data in terms of environmental variation led to hardly any predictions outside the model calibration data (Appendix B, Table B1), minimising potential uncertainties related to extrapolation.


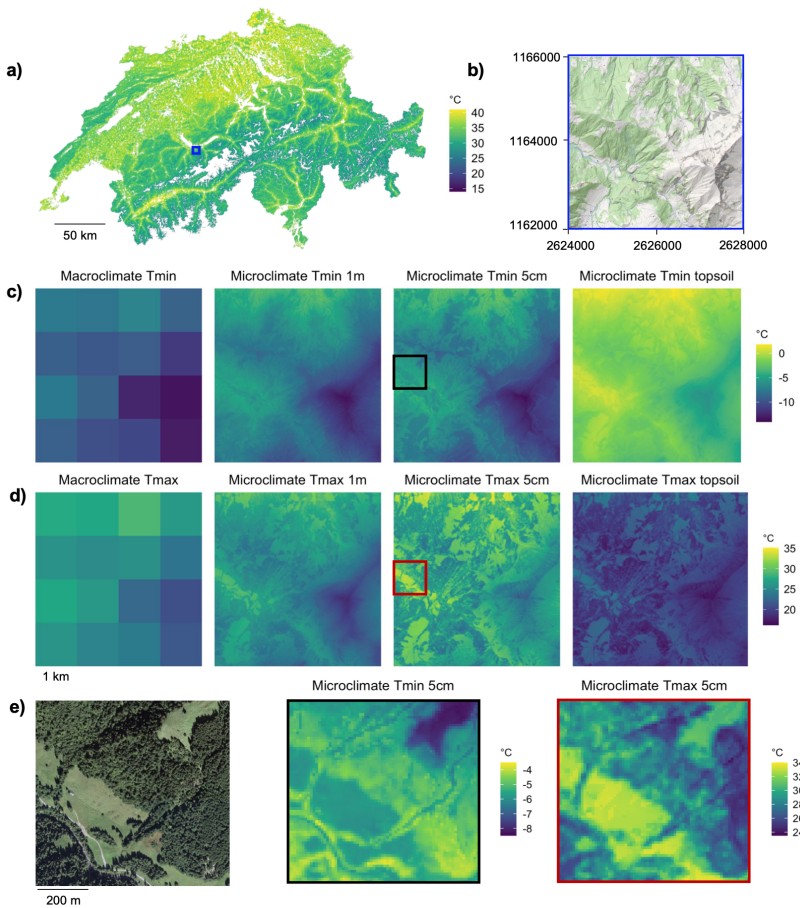

**Figure 6.** Microclimate maps for Switzerland. a) nation-wide map of daily maximum microclimate temperature in summer. b) 4 x 4 km hillshaded sample region with forest cover shown in light green; the location of the region is indicated by a blue rectangle in a). Across the sample region we illustrate our microclimate maps in comparison

with the macroclimate data, with the maps in c) representing the conditions on a frost day in spring, and the maps in d) showing the conditions on a warm summer day. e) shows a smaller area (coloured rectangles in c & d), illustrating the small-scale temperatures during a frost day in April 2021 and a warm summer day in July 2018. Source for macroclimate maps in c) and d): MeteoSwiss.



**4 Discussion**

Our measured microclimate temperatures within an environmentally heterogeneous region revealed strong vertical and horizontal variation in near surface temperatures. This microclimate variation can be mapped with high accuracy at a national scale, thus overcoming a prevalent limitation of macroclimate temperature maps that do not represent small-scale temperature variation. For example, our microclimate mapping approach reveals the distribution of locations that experience substantially reduced daily temperature extremes, such as along forest

density gradients. The temperature buffering effect was particularly pronounced for temperatures in the topsoil 5 cm below ground and air temperatures 5 cm above ground, and less so for air temperatures 1 m above ground, a finding that aligns well with expected vertical temperature profiles in forests (De Frenne et al., 2021). The ability to map this temperature buffering effect along a vertical temperature gradient is expected to provide crucial information to better understand microclimate-species interactions and their implications for biodiversity and

ecosystem functioning (De Frenne et al., 2021; Lembrechts et al., 2019a).

Trees and shrubs have a strong impact on near surface and topsoil temperatures, mainly via their effects on the radiation regime but also via their effects on wind speed and evapotranspiration (Geiger et al., 2009). We found a particularly strong positive effect of direct shortwave radiation on microclimate $T_{max}$ at 5 cm above ground and in the topsoil, implying that incoming radiation is the main controlling variable of ground level microclimates once

the macroclimate is accounted for. In line with the expectation that daily $T_{min}$ are higher under dense canopies because of longwave enhancement (Webster et al., 2016), we found a negative effect of sky view fraction on both above ground $T_{min}$ at both 5 cm and 1 m above ground. Topsoil $T_{min}$ however, was positively affected by sky-view fraction, potentially due to a higher degree of topsoil warming outside forests. These results imply that including high resolution 3D remote sensing data of forest structure and derived radiation estimates significantly increase

our capability to describe microclimatic variation.

Our detailed assessments of radiation effects on microclimatic variation relied on the application of a high-resolution radiative transfer model to estimate controls on radiation below the canopy, considering the position and crown architecture of each tree in the landscape (Webster et al., 2023). Integrating this model into our microclimate mapping approach constitutes an important novelty as it not only furthers more conventional

approaches to estimate vegetation effects on radiation, e.g., via the use of light availability proxies such as canopy height, cover, or leaf area index (LAI), but also provides a pathway to quantify the effect of different natural and management related forest dynamics on near surface and topsoil temperatures. Such analyses now become feasible, as the canopy structure information input into the radiative transfer model can be manipulated to represent past or future forest structure where a subsequent model update would reveal the microclimatic impact quantitatively.

This is particularly relevant because it is increasingly evident that land use effects, e.g., from forest management practices, but also forest disturbances (e.g., due to droughts, bark beetles, wind storms) can have strong immediate effects on microclimate temperatures, invoking microclimate temperature changes that are ecologically more relevant for explaining biodiversity dynamics than macroclimate change (Zellweger et al., 2020; Christiansen et al., 2022). Our approach thus allows for including both past as well as future woody vegetation dynamics, e.g., as

driven by increasing forest disturbances (Senf and Seidl, 2021), into the microclimate modelling, thus addressing an important methodological gap in microclimate science (De Frenne et al., 2021).



We found that microclimate temperatures were considerably reduced at locations with increased topographical wetness and northness, with relatively large effects on $T_{max}$. These effects are expected to be related to generally cooler conditions in north exposed places as well as in places with large lateral, topographically derived water flow, e.g., via higher soil water content and cold air flow. We further found that increases in rain sum, i.e., our variable indicative of soil moisture, resulted in lower $T_{max}$ at 5 cm above ground, confirming previous findings that soil moisture affects near surface microclimates (von Arx et al., 2013). However, our results also show that the overall effect of the rain sum on topsoil and near surface temperature was relatively weak.

The primary output of this work are maps of daily microclimatic temperatures during the vegetation seasons of the years 2012 to 2021. These maps improve some key scale limitations inherent in currently available macroclimate datasets, i.e., they improve on spatial scale by modelling microclimates at 10 m resolution, while maintaining a daily temporal resolution, and they represent microclimates at three vertical heights - 1 m and 5 cm above ground, and 5 cm within the topsoil, thus also improving on vertical resolution. These improvements have a range of implications for future assessments of climate - species interactions, and our understanding of climate change impacts on biodiversity. Species temperature preferences, microclimate heterogeneity, and microclimate refugia, for example, can now be mapped in much greater detail, which is expected to improve the accuracy of analysis and forecasting of species distributions and range dynamics (Lembrechts et al., 2019b; Maclean and Early, 2023). The high resolution of our data also enables a more precise estimation of threshold dependent temperature variables, such as degree days or frost frequencies, which is expected to improve models and approaches that depend on such variables, e.g., models of population dynamics or site suitability assessments for regenerating target tree species. In sum, these maps enable a more realistic, organism-centred perspective when analysing climate-species interactions and are thus relevant to both fundamental and applied ecology, as well as agriculture and forestry in the face of climate change.

**Data and code availability**

The monthly aggregated microclimate maps, as well as the monthly transmissivity maps will be made publicly available on Envidat, a permanent data repository managed by WSL. The daily microclimate maps are freely available upon request. The radiation model CanRad is available on https://github.com/c-webster/CanRad.jl.

**Author's contributions**

FZ designed the study with input from NEZ and PDF. ES, FZ and DF collected the microclimate data. ES, FZ, JTM and CW performed the data analyses and modelling, with input from DNK and TJ. AB and CG provided data. FZ and CW wrote the manuscript. All authors provided feedback to the manuscript.

**Competing interests**

The authors have no competing interests.

**Acknowledgements**

We are grateful to all landowners for granting access and supporting us with the field measurements, and to the Federal Office of Meteorology and Climatology MeteoSwiss for providing macroclimate data. FZ was supported by the Swiss National Science Foundation (Grant Number 193645).



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



**Appendices**

**Appendix A**

**Logger sensitivity analysis**

Ultra-fine wire thermocouples have been shown to accurately measure microclimate air temperatures and have outperformed shielded standard logger types in measuring air temperatures in locations exposed to direct sunlight or close to the ground, of which our sampling design contained many (Maclean et al., 2021). To test potential

effects of direct solar radiation on the thermocouples measurements we performed a shielding experiment. This experiment consisted of a paired design, where we placed two shielded and two unshielded thermocouples in each of three representative environments for the field sampling (open, trees outside forest, and forest). The shields consisted of a lid of aluminium foil such that the thermocouple was just shaded but not isolated from wind (Appendix E, Fig. E1). The experiment was carried out during sunny conditions over three weeks in summer 2022.

The analysis revealed an overall RMSE of 0.35 °C for the daily maximum temperatures between the shielded and unshielded thermocouples, a value that is close to the measurement accuracy of 0.3° C. We also did not find a significant difference between the three environments tested. This sensitivity analysis confirmed previous findings (Maclean et al., 2021) and shows that the effect of direct solar radiation on the thermocouples is negligible in our study set-up.




**Appendix B**

**Table B1**. Comparison of sampled predictor variable range and the observed predictor variable range across the entire area to which predictions were made. Sampled range describes the predictor variables extracted at sampling plot locations; predicted range are the 1st and 99th percentiles of a random subset (n = 10'000) of observed values across the predicted microclimate maps. Note that we did not include the macroclimate and rainsum variables in this table because they are dynamic variables with a daily resolution in our models, yet as our sampled regions cover the observed macroclimate temperature and rainfall patterns across Switzerland well, we are confident that the sampled range matches the predicted range.

| Variable name | Sampled range (mean) | Predicted range (mean) | Unit |
|---|---|---|---|
| Direct radiation proxy | 0 – 29410 (8600) | 0 – 29540 (18570) | kJ/m2/day |
| Skyview fraction | 2 – 98 (41.5) | 4 – 99 (62.7) | % |
| Vegetation height | 0 – 35.1 (13.3) | 0 – 37.5 (7.4) | m |
| Topographic position | -1.0 – 0.97 (-0.04) | -1.7 – 1.7 (0.0) | Index |
| Topographic wetness | 1.1 – 9.6 (3.2) | 0.5 – 10.5 (3.3) | Index |
| Northness | -1 – 1 (-0.1) | -1 – 1 (0.0) | Index |
| Slope | 0.3 – 41.7 (17.9) | 0.3 – 59.3 (20.8) | Degrees |



**Appendix C**

**Table C1**. Descriptive statistics for temperature offsets (°C). The values indicate the range (and mean between brackets) calculated by deducting the microclimate temperature from the lapse rate corrected macroclimate temperature. Negative offset values thus indicate cooler microclimates compared to the macroclimate, and vice-versa. All means were significantly (p < 0.01) different from zero.


| | Tmax 1m | Tmax 5cm | Tmax soil | Tmean 1m | Tmean 5cm | Tmean soil | Tmin 1m | Tmin 5cm | Tmin soil |
|---|---|---|---|---|---|---|---|---|---|
| Forest | -14.0 – 11.4 (-1.3) | -14.7 – 26.2 (-1.1) | -16.5 – 8.7 (-5.2) | -6.5 – 6.4 (-0.2) | -7.1 – 5.1 (-0.4) | -10.4 – 7.4 (-1.5) | -6.0 – 10.9 (1.8) | -9.4 – 10.1 (1.7) | -5.7 – 11.6 (2.6) |
| Trees outside forests | -13.9 – 6.7 (-0.7) | -11.7 – 27.8 (-0.2) | -15.5 – 6.9 (-4.9) | -7.5 – 6.3 (-0.3) | -7.9 – 16.0 (-0.6) | -11.6 – 7.6 (-1.3) | -5.4 – 7.1 (1.1) | -8.8 – 11.1 (0.9) | -6.4 – 11.1 (2.3) |
| Open | -15.2 – 8.3 (-0.1) | -17.0 – 20.7 (3.4) | -18.3 – 17.8 (-0.9) | -8.6 – 5.3 (-0.2) | -12.2 – 6.8 (-1.0) | 13.3 – 8.7 (0.6) | -6.1 – 7.1 (1.0) | -11.7 – 8.0 (-0.5) | -6.9 – 12.6 (3.5) |
| Overall | -15.2 – 11.4 (-0.9) | -17.0 – 27.8 (0.1) | -18.3 – 17.8 (-4.2) | -8.6 – 6.4 (-0.2) | -12.2 – 16.0 (-0.6) | -13.3 – 8.7 (-1.0) | -6.1 – 10.9 (1.5) | -11.7 – 11.1 (1.1) | -6.9 – 12.6 (2.7) |



**Appendix D**

**Table D1.** Predictive performance of microclimate models as quantified using block cross validation. We also report the predicted offset ranges, calculated as the difference between the predicted microclimate and the macroclimate.

| Model | | Linear mixed effects model | | | Random Forest | | | GAMM | | |
|---|---|---|---|---|---|---|---|---|---|---|
| | | R2 | RMSE | Predicted offset range | R2 | RMSE | Predicted offset range | R2 | RMSE | Predicted offset range |
| **Temperature 1 m above ground** | $T_{max\ micro}$ | 0.92 | 1.7 | -3.0 – 1.4 (-0.6) | 0.90 | 2.2 | -14.1 – 15.2 (-1.1) | 0.87 | 2.4 | -4.7 – 10.3 (-0.3) |
| | $T_{mean\ micro}$ | 0.95 | 1.2 | -2.4 – 1.9 (-0.1) | 0.92 | 1.8 | -12.3 – 12.2 (-0.4) | 0.93 | 1.4 | -3.0 – 3.9 (0.6) |
| | $T_{min\ micro}$ | 0.92 | 1.5 | -0.8 – 3.4 (1.5) | 0.89 | 1.9 | -8.5 – 13.2 (1.2) | 0.88 | 1.8 | -3.3 – 9.8 (1.6) |
| **Temperature 5 cm above ground** | $T_{max\ micro}$ | 0.73 | 3.4 | -5.6 – 7.0 (0.4) | 0.72 | 3.6 | -14.9 – 14.6 (-0.1) | 0.53 | 5.2 | -8.4 – 17.3 (1.4) |
| | $T_{mean\ micro}$ | 0.9 | 1.6 | -3.3 – 3.6 (-0.4) | 0.86 | 2.0 | -13.9 – 12.5 (-0.8) | 0.83 | 2.1 | -8.3 – 7.5 (-0.6) |
| | $T_{min\ micro}$ | 0.88 | 1.8 | -1.3 – 4.3 (1.2) | 0.81 | 2.4 | -10.2 – 14.8 (0.8) | 0.32 | 6.2 | -5.1 – 16.4 (0.5) |
| **Temperature 5 cm below ground** | $T_{max\ micro}$ | 0.54 | 3.1 | -15.4 – 8.0 (-4.0) | 0.63 | 2.9 | -21.1 – 15.2 (-4.4) | 0.26 | 5.1 | -34.9 – 24.9 (-3.3) |
| | $T_{mean\ micro}$ | 0.75 | 2.0 | -7.5 – 9.1 (-0.9) | 0.74 | 2.2 | -13.4 – 17.0 (-1.2) | 0.55 | 2.9 | -10.9 – 17.6 (-0.5) |
| | $T_{min\ micro}$ | 0.78 | 1.9 | -1.5 – 9.5 (2.9) | 0.73 | 2.1 | -9.5 – 16.7 (2.5) | 0.56 | 2.8 | -8.1 – 23.4 (3.1) |





**Appendix E**

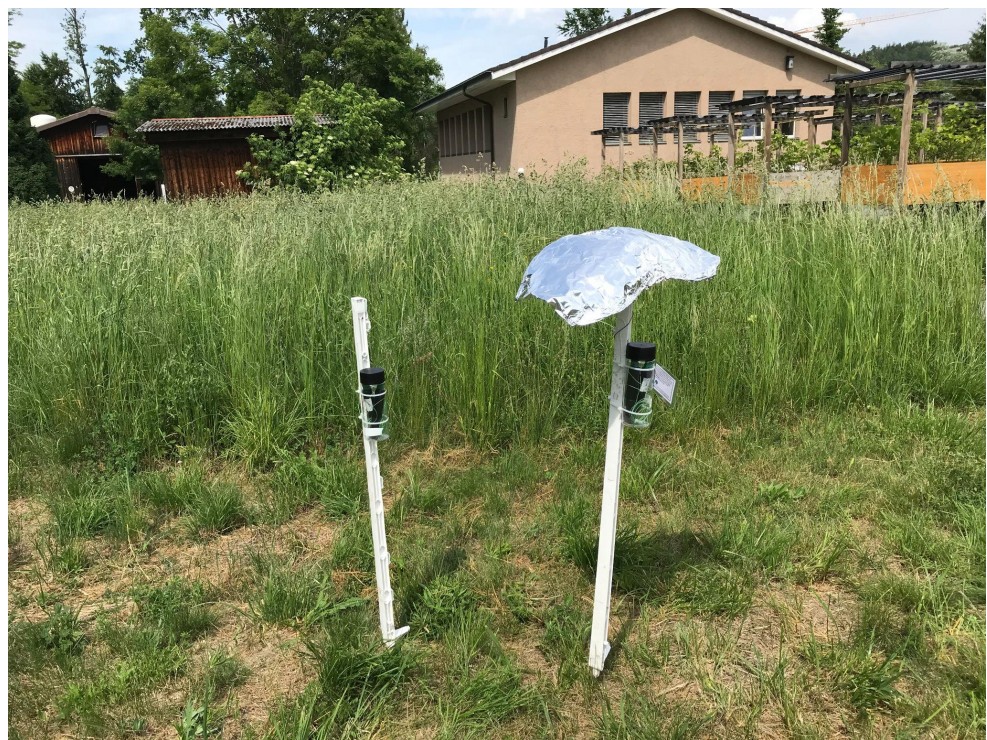

**Figure E1**: Example of shielding experiment with aluminium foil to determine if the logger measurements were
affected by direct sunlight on the thermocouple. The thermocouple on the right pole was placed such that it was
shaded throughout the day. The actual experiment consisted of a paired design, where we placed two shielded and
two unshielded thermocouples in each of three representative environments for the field sampling (open, trees
outside forests, and forest).



**Appendix F** 705

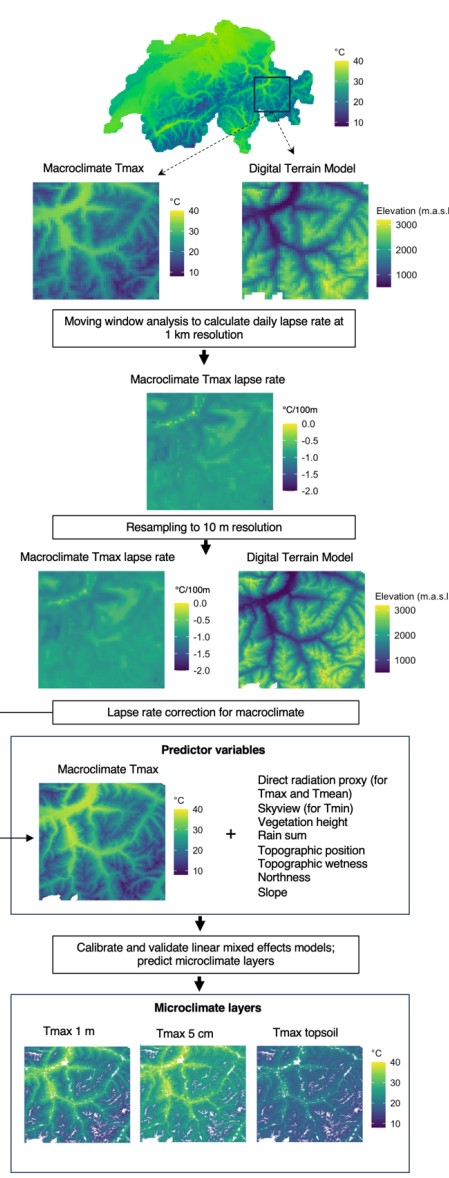

**Figure F1**: Flowchart describing the data processing and modelling pipeline. We used daily macroclimate rasters and a digital terrain model at 1 km resolutions to calculate daily lapse rate maps, which were resampled to 10 m resolution and used to apply a lapse rate correction to our macroclimate layers. Together with other predictor variables we used the lapse rate corrected macroclimate layers to predict our microclimate layers at different vertical heights. Please note that for illustration purposes all temperature maps show daily maximum temperatures prevailing on the 31st of July 2018.