# Peer review of "Microclimate mapping using novel radiative transfer modelling"

_EGUsphere, 2023_

## Author Response (AR1)

**Manuscript EGUSPHERE-2023-1549t**

**Zellweger et al. Microclimate mapping using novel radiative transfer modelling**

**Response to comments by reviewers and peers**

**Reviewer 1**

Zellweger et al.: We would like to thank Jonas Lembrechts for his positive and constructive review.

"This is a very elegant microclimate modelling exercise, well written and explained, and thoroughly done. The authors have a good dataset, well-thought-through explanatory variables and a waterproof modelling approach, and the end-result are very useful high-resolution (spatially and temporally) maps of microclimate across the whole of Switzerland.

In my opinion, this is how microclimate modelling should be done! I have very few comments and questions. Some of the questions I THINK I know the answer for, but it would make things easier for me as a reader if they would be spelled out in the text.

I am very happy with the way radiation was treated in this analysis, showing us a way to a future with more accurate microclimate models at large scales.

I also applaud the use of thermocouples for aboveground temperature measurements. This is the first in a long series of reviews I made where aboveground temperature was measured with thermocouples, and where I did not have to warn for the risks associated with low-cost sensor measurements under direct sunlight. In that regard, your comparative test (Appendix A) is awesome and very useful. However, I wonder if comparing only the daily maxima is sufficient. If at any point sun would reach the thermocouple and heat it, you could be observing a sudden unexpected peak. As I only have that value and nothing more to base myself on, I can't judge of something like that happened. What about also providing RMSE for daily mean and perhaps hourly-level RMSEs?"

Zellweger et al.: Thank you for your encouraging comments. We have checked the raw data of our sensitivity analysis in Appendix A again and found that the RMSE for daily mean temperatures are even lower (i.e., 0.18 °C) than for the daily maxima that werereported in the manuscript (i.e., 0.35 °C), providing further evidence that our measurements are not affected by direct sunlight. We have added the new RMSE to the manuscript.

"L36-37: more biologically relevant than what?"

Zellweger et al.: We were referring to commonly used macroclimate data but acknowledge that the wording was unclear, and have adjusted it accordingly.

"L129: is there a maximum width for a hedge, after which you would consider it a forest? Also, could you specify how many sensors under hedges and how many sensors under single trees you had?"

Zellweger et al.: We considered a hedge to have a maximum width of 5m. Out of the 22 plots below trees outside forests, 10 were below solitary trees or small isolated groups of trees and 12 within hedge type habitats.

"L134-135: a bit clunky formulated: does this mean forest plots were only sampled in these two regions (I guess not)? And what are the different forest types there? Is that point on forest types relevant if what you actually say is that you did not sample outside of forests there? Does the 'as a result' on L136 refer to the lack of non-forested plots in those two regions, as this is counterintuitive then. I think in this part you are trying a bit too hard to justify a limitation of the design, creating confusion in the process. I suggest to shorten this and just state it dryly."

Zellweger et al.: Thank you for this comment. We agree and have rephrased these two sentences.

"L158: also cool: georeferencing at a 1 m resolution – much needed for high-resolution microclimate modelling in the future!

L161: the longest vegetation period – I guess that only refers to 2022?"

Zellweger et al.: Yes indeed, we have slightly adjusted the wording to be more clear about this.

"L161: could you say 'the few remaining snow days within the April-October period'? Or something? My main question is in fact – was there snow between April and October and if so, how much, but I don't think we need exact numbers."

Zellweger et al.: Snow was observed on c. 10 plots at high elevations during some days mainly in April 2022. We carefully checked all microclimate temperature time series to identify the respective periods and removed them from the analysis.

"L278: what is meant with 'local' catchment and 'local' slope? Is this within a radius or everything that's higher up? I assume local means something different for catchment and slope?"

Zellweger et al.: The "local" refers to the focal DTM pixel location. We have now simplified the wording.

"L290: you say 'spatial predictor variables', but isn't macroclimate temperature (and rain sum) spatiotemporal (and isn't it key to have a temporal component in the model if you pool all daily data? Or did you make separate models for each day? If the former, wouldn't we need a DOY-parameter to correct for seasonal effects?"

Zellweger et al.: The word "spatial" was somewhat misleading as indeed many of our predictor variables are spatiotemporal. We have adjusted the wording now.

"L294: I understand you don't use the offsets for modelling, but I have a hard time finding in the methods what you DO use the offsets for. Or am I overlooking things?"

Zellweger et al.: Yes, the offsets were used to describe the differences between the microclimate and macroclimate data, i.e. the microclimate variation not captured in macroclimate data. We have now adjusted the text at the end of section 2.2 to be more clear about this, and also mention it again in section 2.4. We used the actual plot-level microclimate measurements as dependent variables because this allowed us to predict maps of observed temperature ranges that can more directly be used to study species-climate relationships than the temperature offsets that would first need to be converted to actual temperatures.

"L305: are these two-variable splits and 500 trees following conventions? Best to add if so!"

Zellweger et al.: Thanks, we have added this point.

"L316: comparison with which metric?"

Zellweger et al.: We compared R2 values and root mean squared errors (RMSE) and have now also added this information here.

"L330: could you quantify 'particularly high' here in the text, e.g. using the standard deviation or 5-95% interval? Just for the ease of reading."

Zellweger et al.: The 5th and 95th percentiles are now mentioned.

"L478: perhaps still worth mentioning that while the daily resolution is awesome, the next step would be to have predictor variables such as forest cover temporally explicit as well?"

Zellweger et al.: Indeed. This comment is much related to our discussion paragraph about including vegetation dynamics into the microclimate modelling where we outline potential future avenues for microclimate modelling based on temporally dynamic data about forest structure and cover. Following your comment we have added a sentence at the beginning of this paragraph.

"L480: these are monthly maps for each of the years between 2012 and 2021, right, and not averaged across that decadal period? Just to be sure…"

Zellweger et al.: Yes, in fact, we provide all the different aggregation steps separately if needed.

"I really love Table B1! This is an excellent example of the point I aim to make repeatedly, that with relatively few, smartly located sensors, a large area can be covered accurately.

Here and there some language errors (e.g., L115: 4% of the country is), I am assuming someone else will catch these in the next steps"

Zellweger et al.: Thanks for spotting this.

"Table D1: I have a very hard time understanding how the offset ranges can be so massively different (around -2 versus around -10 and more, roughly speaking) between the different models. Shouldn't they all have more or less the same offsets if the predictions are the same. Or does that mean predictions are around 10°C different for the different methods? That'd be super worrisome, no?"

Zellweger et al.: The predicted offset ranges in Table D1 are based on the difference between the predicted microclimate temperatures and the macroclimate temperatures. While it is true that the minimum and maximum predicted offsets vary considerably between the three modelling approaches, the mean predicted offsets are actually very similar. The overall predicted offsets do not vary by more than a few decimal degrees among the three modelling approaches (except with GAMM for Tmax micro where the difference is about a degree C). However, it becomes evident that Random Forest predictions result in a larger variability in predicted offsets compared to LMEs and GAMMs, but the model performance based on block cross-validation suggests that LMEs generally perform better or as good as Random Forests or GAMM. We hope this clarifies the interpretation of Table D1.

"Figure 4: these are daily observed and predicted temperatures? Is there a correction for spatial and temporal autocorrelation needed (if you have a large amount of days at the same location, these will be strongly correlated and might inflate the models, for example. Or, are these separate models for each day? See my question above!)"

Zellweger et al.: Yes these are the daily observed and predicted temperatures, based on one model (not separate models for each day). To account for the pseudoreplication within regions and associated spatial autocorrelation we added the region as a random effect term, however, we did not specifically account for temporal autocorrelation in the modelling.

Reviewer 2

"I found the manuscript to be interesting and extremely rigorous but the Discussion a bit brief. It is clear that the approach accurately captures many aspects of microclimate and the uncertainty analysis is quite rigorous, but going forward what data inputs or modeling advancements are necessary to further improve the analysis? This forward-looking context was missing to me and I feel that the manuscript could be improved by doing so."

Zellweger et al.: Thank you for this positive review. We agree that a forward-looking perspective should be provided and have therefore extended our paragraphs outlining potential avenues for dynamic microclimate modelling based on dynamics in vegetation cover and structure. In the last paragraph we also write beyond the more technical avenues for future microclimate modelling and state some of the implications of microclimate data for future ecological research.

**CC1**

This is a useful contribution the microclimate modelling and mapping literature and, overall, a well-executed paper. However, the framing could be sharpened in a few places:
> The need for microclimate data is presented primarily as a scale issue, but it is not just that. There are consistent differences between near-ground / near vegetation / below canopy climatic conditions and macroclimate irrespective of scale. The manuscript presents high-resolution estimates of temperatures at multiple heights, so arguably has two benefits.

Zellweger et al.: Thank you for pointing this out. We entirely agree and have adjusted our original wording in the introduction and discussion where we mention these benefits.

In stating that a remaining challenge in microclimate mapping is incorporating radiation transfer through vegetation canopies, you somewhat simplify what existing microclimate models do. The mentioned models do account for solar angles and relative diffuse / direct fraction in their published form, and in the newest form on Github include a two-stream radiative transfer model. They don't do is use a finite element approach – interactions with directional radiation are instead handled by characterising the foliage as having a continuous distribution of inclination angles. I think the CanRad model used in this paper is a finite-element model (though this is not explicitly stated), so there are benefits to using it. However, these benefits may be lost as when predicting microclimate across Switzerland, you use simpler proxies that may not be any better than those included in existing models. It would be helpful to present more clearly what the particular benefits of your approach are.

Zellweger et al.: This is well observed and we regret that we have not explicitly stated and referenced some the latest developments in mechanistic microclimate modelling, e.g. Maclean & Klinges (2021). This shortcoming, which was also pointed out by Michael Kearney (CC2 below), has now been corrected.

Additionally, Fig. 4 appears to show very good correspondence between observations and predictions, which is encouraging. However, I was left wondering how much of this was because most of the variance is driven by macroclimate. What happens if you plot the observed and predicted offsets against one another?

Zellweger et al.: We agree. As stated in our manuscript the largest part of the variance is driven by the macroclimate. This is also expected considering that our study area covers large macroclimatic gradients in both space (e.g., the plot elevations ranged from 195 to 2463 m.a.s.l., with mean macroclimate temperatures often ranging more than 14 °C on a single day) and time (i.e, the entire vegetation period). A detailed cross-validation analysis confirms this expectation, suggesting consistent but often minor increases in predictive performance of the microclimate model (Appendix E, now included in the revised manuscript). It is worth noting that in spite of these relatively minor increases in statistical accuracy, a key advantage of our microclimate model is that it reveals the small scale variability in near surface temperature conditions, e.g. a 5°C difference in Tmax 5 cm above ground over a 50m gradient from outside to inside a forest, whereas the macroclimate model predicts a uniform and (slightly) less accurate temperature value over the same distance, being insensitive to local drivers of microclimate and its implications for biodiversity and ecosystem functioning. We suggest keeping the actual temperature values in Fig. 4 as those are the dependent variables in our modelling, not the temperature offsets.

CC2:
This is nice work. Note that Maclean and Klinges's microclimc package has algorithms for dealing with solar radiation through canopies and attenuation of wind speed. It might also be interesting to see how well point-based models that incorporate physical processes more explicitly (e.g. heat flow through heterogeneous soils, soil moisture) compare. The NicheMapR microclimate model (see Kearney and Porter, 2017 and Kearney et al., 2020) is such a point model and can take your sky-view factors into account for longwave radiation as well as take your adjusted solar radiation as a direct input. It doesn't incorporate lateral movement of heat which your statistical approach does. It would be interesting to see how well each approach does for time series at a point given the different emphases on statistical and physical depiction.

Kearney, M. R., & Porter, W. P. (2017). NicheMapR - an R package for biophysical modelling: The microclimate model. *Ecography*, *40*(5), 664–674. https://doi.org/10.1111/ecog.02360

Kearney, M. R., Gillingham, P. K., Bramer, I., Duffy, J. P., & Maclean, I. M. D. (2020). A method for computing hourly, historical, terrain-corrected microclimate anywhere on earth. *Methods in Ecology and Evolution*, *11*(1), 38–43. https://doi.org/10.1111/2041-210X.13330

Maclean, I. M. D., & Klinges, D. H. (2021). Microclimc: A mechanistic model of above, below and within-canopy microclimate. *Ecological Modelling*, *451*, 109567. https://doi.org/10.1016/j.ecolmodel.2021.109567

Zellweger et al.: This feedback is much appreciated. As stated in the responses below we now refer to Maclean and Klinges's microclimc package and, in combination with remarks from Reviewer 2, have extended the discussion section to reflect these inputs and references.